Genomic evidence of adaptive evolution in the reptilian SOCS gene family

http://orcid.org/0000-0002-0097-6949 Xia Tian
Zhang Lei
Sun Guolei
http://orcid.org/0000-0001-7521-954X Yang Xiufeng
Zhang Honghai zhanghonghai67@126.com
College of Life Science, Qufu Normal University , Qufu, Shandong , China
Lahr Daniel
Electronic publication date: 2021 Jun 24
Publication date: 2021
Volume: 9
Electronic Location ID: e11677
Received 2020 Dec 2; Accepted 2021 Jun 4
Copyright: © 2021 Xia et al.
Copyright year: 2021
Copyright holder: Xia et al.
License: This is an open access article distributed under the terms of the Creative Commons Attribution License, which permits unrestricted use, distribution, reproduction and adaptation in any medium and for any purpose provided that it is properly attributed. For attribution, the original author(s), title, publication source (PeerJ) and either DOI or URL of the article must be cited.
License URL: https://creativecommons.org/licenses/by/4.0/

Keywords: SOCS gene family, PAML, Reptiles, Adaptive evolution

Funding: National Natural Science Foundation of China 31872242 and 31672313 Forest Scientific Research in the Public Welfare 201404420 This work was supported by the National Natural Science Foundation of China (Nos. 31872242 and 31672313) and the Special Fund for Forest Scientific Research in the Public Welfare (No. 201404420). The funders had no role in study design, data collection and analysis, decision to publish, or preparation of the manuscript.

==============================
The suppressor of the cytokine signaling (SOCS) family of proteins play an essential role in inhibiting cytokine receptor signaling by regulating immune signal pathways. Although SOCS gene functions have been examined extensively, no comprehensive study has been performed on this gene family’s molecular evolution in reptiles. In this study, we identified eight canonical SOCS genes using recently-published reptilian genomes. We used phylogenetic analysis to determine that the SOCS genes had highly conserved evolutionary dynamics that we classified into two types. We identified positive SOCS4 selection signals in whole reptile lineages and SOCS2 selection signals in the crocodilian lineage. Selective pressure analyses using the branch model and Z-test revealed that these genes were under different negative selection pressures compared to reptile lineages. We also concluded that the nature of selection pressure varies across different reptile lineages on SOCS3, and the crocodilian lineage has experienced rapid evolution. Our results may provide a theoretical foundation for further analyses of reptilian SOCS genes’ functional and molecular mechanisms, as well as their roles in reptile growth and development.

Introduction

Cytokines are multifunctional proteins and essential intercellular regulators that are involved in innate and adaptive inflammatory organism defense, cell development, and repair processes via different signaling mechanisms (Oppenheim, 2001). Suppressors of cytokine signaling (SOCS) are some of the most crucial feedback inhibitors in the prevention of excessive cytokine signaling and maintenance of homeostasis and normal cellular functions (Hong-Jian et al., 2008). The SOCS proteins function as negative feedback inhibitors, controlling particular cytokine signals in order to regulate cellular responses and maintain a stable environment (Linossi, Calleja & Nicholson, 2018). During the phosphorylation of signal transducer and activator of transcription (STAT) proteins, SOCS proteins helps regulate various cytokines by combining the kinase inhibitory region (KIR) and members of the Janus kinase (JAK) family (Tannahill et al., 2005). The SOCS gene family was initially identified in mammals and was comprised of eight members, including SOCS1–7 and the cytokine-inducible SH2-containing protein (CISH) (Linossi & Nicholson, 2015). Additional studies found that all SOCS family members shared a conserved structure: an N-terminal domain, a highly conserved C-terminal motif (called the SOCS box), and a central SH2 domain (Hao & Sun, 2016; Krebs & Hilton, 2001). Today, the SOCS family of proteins is further classified based on amino acid residue into type I (SOCS4–SOCS7) and type II (SOCS1–SOCS3, and CISH). A type I subfamily member contains an extensive N-terminal region compared to a type II subfamily member (Jin et al., 2008).

Previous studies focused on the SOCS gene mechanisms associated with severe diseases, such as asthma, atopic dermatitis, and lymphoma (Weniger et al., 2006; Yoh-ichi et al., 2003). SOCS1 is expressed in melanoma cell line masses and is related to tumor invasion, stages of disease, and thickness. Therefore, SOCS1 can be considered a therapeutic target for cancer (Scutti et al., 2011). SOCS3 is especially important in the development of leptin resistance, whereas SOCS1, SOCS3, SOCS6, and SOCS7 can reduce insulin activity (Howard & Flier, 2006). Notably, silencing SOCS gene expression when it is up-regulated by interferon negative regulators (IFN) is an effective strategy in increasing the antitumor effect of IFN (Takahashi et al., 2008). The immunomodulatory effects of SOCS proteins suggest that natural killer cells are their potential targets, providing a basis for novel cancer therapies (Keating & Nicholson, 2019). Additionally, a previous study found that imbalance among heterogeneous SOCS proteins may result in multiple sclerosis pathophysiology (Toghi et al., 2017).

The significance of SOCS regulation in immune and other essential cellular responses has been confirmed using SOCS-deficient mice (Alexander et al., 1999; Boyle & Robb, 2008; Greenhalgh et al., 2002). Their evolution has been previously studied in fish and other vertebrates (Hong-Jian et al., 2008; Tiehui et al., 2010; Wang et al., 2019). A series of functional studies on the eight SOCS gene family members have also been performed in mammals (Linossi & Nicholson, 2015). Although SOCS gene family members have been identified and characterized in mammals and several other species, far less is known about the evolutionary patterns of SOCS genes in reptiles, and the evolutionary relationship and orthology of SOCS genes in reptiles has remained unexplored. Originating more than 250 million years ago (Reisz, Modesto & Scott, 2011), reptiles have continued to occupy a significant position in natural systems, serving as essential bioindicators for ecological environments and a basis of research on the biological evolutionary process during speciation (Raxworthy et al., 2010). An understanding of reptilian SOCS family evolution will provide meaningful insights to the evolutionary history of reptilian immunity. Previous studies have detected that the SOCS gene repertoire differs across species (Wang et al., 2019). For example, researchers failed to identify the SOCS homologue in ctenophores and choanoflagellates, but could identify homologues of six SOCS gene members in Porifera (Liongue, Taznin & Ward, 2016). However, recent studies on reptiles have shown that their immune systems tend to have similar components to their mammalian counterparts with subtle differences (Zimmerman, 2020). We hypothesized that, similar to mammals, the SOCS family of reptiles followed classical expansion during the two rounds of whole-genome duplication with intact SOCS family members. Due to the current lack of extensive evidence on SOCS family molecular evolution across the reptilian phylogeny, we implemented evolutionary analysis on reptiles’ whole-genome sequences in this study. As more whole-genome data becomes available for a greater number of reptiles, the evolutionary and structural characteristics of reptile SOCS genes has become more attainable. The SOCS sequences extracted from reptilian genomes and those from public resources provided us with an excellent opportunity to explore reptile evolutionary selection diversity. Reptiles, the only ectothermic amniotes, have wide ranges of habitat, modes of diet, behaviors, lifespans, and reproduction (Zimmerman, 2020). It has been demonstrated that reptile body temperature cannot be kept constant and will undergo seasonal shifts with environmental temperature, and infection is strongly related to body temperature (Zimmerman, Vogel & Bowden, 2010). A previous study found that the SOCS family might have adapted to natural environmental changes (Tian et al., 2020). In this study, we investigated the evolution of SOCS genes in reptiles and detected the evolutionary selection diversity in reptile lineage types. We aimed to test our hypothesis that these SOCS genes are under adaptive evolution across reptiles to determine if different reptilian clades experienced different selective regimes.

Materials & methods

Species and sequences

In this study, we chose 23 reptilian genomes to extract eight SOCS genes downloaded from the National Center for Biotechnology Information (NCBI). The reptilian genome information is summarized in Table S1. We obtained and directly downloaded the previously sequenced SOCS genes for several reptilian species using an online Web BLAST search and the NCBI database. We set these sequences as queries to explore reptile genomes that have not been annotated. We created a local database for each reptilian genome and used the BLASTN and TBLASTN in BLAST v2.7.1 to search SOCS encoding sequences. We used an e-value of 10−5 as the default cut-off to confirm significant matches against the genome. SOCS sequences confirmed by the BLAST searches were applied in reciprocal balstx searches of human proteomes to improve the accuracy of the ortholog matches. The SOCS gene sequences and accession numbers are shown in Table S2.

Phylogenetic analysis of reptile SOCS genes

The 260 reptilian SOCS gene sequences were aligned based on their amino acid translations using Multiple Sequence Comparison by Log-Expectation (MUSCLE v3.8.31) (Edgar, 2004). The phylogenetic relationship of the reptile SOCS genes was established using RaxML v8.2.12 with 1,000 bootstrap replications. We applied jModelTest (Posada, 2008) with Akaike Information Criterion (AIC) to test the most suitable nucleotide substitution model, and determined that the GTR + Γ model was the most appropriate for detecting the evolutionary relationship across the eight SOCS gene family members. Finally, we used the iTOL online software (http://itol.embl.de) to visualize and beautify the phylogenetic tree.

Evolutionary pressure analysis

Positive Darwinian selection pressures acting on genes are usually determined by calculating the nonsynonymous (dN)/synonymous (dS) substitution ratio (ω) between homologous protein-coding gene sequences. Very simply, ω (evolutionary rate) >1, <1, and =1 represent positive selection, negative selection, and neutral evolution, respectively. The estimated that the ω ratio between homologous protein-coding sequences is a powerful symbol of positive selection at the molecular level. Phylogenetic Analysis by Maximum likelihood (PAML) is a program package used for phylogeny-based analysis to estimate molecular evolution, and its program CODEML determines positive and purifying selection sites or branches (Yang, 2007). Based on the sequence alignments and phylogenetic trees downloaded from TimeTree (http://www.timetree.org/), we carried out the selective force imposed on reptilian SOCS genes using a codon-based codeml PAML 4.9 d program (Yang, 2007). Several analyses were implemented to test the hypothesis that SOCS genes experienced natural selection across reptile species. To determine the signatures of natural selection on SOCS genes in extant reptiles, we used the site model in codeml to explain the different functions and structure constraints undergone by amino acid sites. The M7 (model = 0, NSsites = 7)/M8 (model = 0, NSsites = 8) pair of codon-based models, which allowed the ω to vary across sites but not across lineages, was included in the site model. This pair had twice the difference compared to the log-likelihood values, and we applied a Chi-squared distribution to estimate the significance. The posterior probabilities (PP) were calculated using empirical Bayes analysis of positive selection sites in the M8 model. Additionally, we used the fixed-effect likelihood (FEL), single likelihood ancestor counting (SLAC), and mixed-effects model of evolution (MEME) in HyPhy (Pond & Frost, 2005) to detect the positive selection sites in the Datamonkey web server. Sites with PP > 0.95 for the M8 model, and a P-value < 0.1 for the FEL, SLAC, and MEME models were considered to have undergone positive selection. HyPhy software packages provided better analysis power and additional advantages for our study compared to PAML (Bulmer & Crozier, 2006; Kosakovsky Pond & Frost, 2005).

We calculated the entire mean rate of dN and dS substitutions in the SOCS coding sequences using the Z-test of selection in MEGA 5.2, the Nei and Gojobori method with Jukes–Cantor correction, 95% site coverage cut-off, and 1,000 bootstrap replicates. We then utilized branch models in the codeml program to determine whether there were differences in the selective forces acting on SOCS genes in diverse reptilian lineages. The branch model permits variable ω ratios across lineages, but changeless ω ratios in the sites, and it could be applied to detect changes in selective pressures in specific branches (Yang & Nielsen, 2002). Several targeted branches were set as one foreground branch, and the others were assigned as background branches. For this, we processed a null one-ratio model (model = 0, NSsites = 0) that estimated the same ω for all branches against the two-ratio model (model = 2, NSsites = 0) that estimated a variable ω in a specific branch using a likelihood ratio test (LRT). A P-value < 0.05 was selected to reject the null one-ratio model and evaluate the significance of the alternative hypothesis. In order to further compare the evolutionary rates of the eight SOCS genes in response to divergent reptile clades, we used the Clade model C (CmC, model = 3, NSsites = 2), which allows codon sites to evolve discrepantly along with the clade (Baker et al., 2016). The intensity of CmC selection was permitted to differ across clades through the use of a different ω for each clade. SOCS gene sites that had undergone positive selection along each branch were further identified using the branch-site model. The branch-site model identified positive selection at specific sites along the specific lineages (Zhang, Nielsen & Yang, 2005) and on a few sites in a few branches. Finally, the likelihood ratio test was utilized to contrast a null model Ma0 (model = 2, NSsites = 2, fix-omega = 1, omega = 1) and a model Ma (model = 2, NSsites = 2, fix-omega = 0, omega = 1) of positive selection pressures on the foreground branch.

Recombination and motif composition analysis

Recombination analysis of eight SOCS gene encoding sequences was performed using GARD in the Datamonkey web server. The recombination of genes may mislead the phylogenetic estimation process and distort following inferences based on inferred phylogenesis, and there will be a high false-positive rate when the sequence being analyzed undergoes recombination (Kosakovsky Pond et al., 2006). The maximum χ2 method was employed to estimate the likelihood of recombination events and to explore putative break-points within SOCS genes. Conserved motifs analysis was performed using the Multiple Expectation Maximization for Motif Elucidation (MEME) online program (http://meme-suite.org/) to obtain information about the similarity and motif distribution of SOCS genes. The parameters applied in the analysis were as follows: minimum motif width = 6 and maximum motif width = 200.

Results

SOCS gene sequences

In this study, we downloaded all available SOCS genes from 16 reptiles, identified or predicted, from GenBank (http://www.ncbi.nlm.nih.gov/genbank/). The reptilian genomes were obtained from NCBI (https://www.ncbi.nlm.nih.gov/genome/) and included Testudines (Apalone spinifera, Malaclemys terrapin, Cuora mccordi, Chelonoidis abingdonii, Gopherus agassizii, Platysternon megacephalum, Malaclemys terrapin, Cuora mccordi, Chelonoidis abingdonii, Gopherus agassizii, Platysternon megacephalum), Serpentes (Crotalus viridis, Vipera berus, Crotalus horridus, Protobothrops flavoviridis, Pantherophis guttatus, Ophiophagus hannah, Crotalus pyrrhus, Hydrophis cyanocinctus, Hydrophis hardwickii, Thermophis baileyi), Sauria (Lacerta viridis, Lacerta bilineata, Paroedura picta, Podarcis muralis, Sphenodon punctatus), and Crocodilia (Alligator mississippiensis, Gavialis gangeticus). The available information was integrated (Table S1). SOCS gene repertoires from some vertebrates have previously been described and used as query sequences to screen orthologs in reptilian genomes. We identified several partial reptile SOCS gene sequences and then discarded these genes for the following analysis. As a result, a set of 260 SOCS gene sequences with intact structures were collected from the reptiles: 30 CISH, 33 SOCS1, 29 SOCS2, 33 SOCS3, 37 SOCS4, 38 SOCS5, 38 SOCS6, and 22 SOCS7 sequences. The SOCS gene family sequences are available on figshare (DOI: 10.6084/m9.figshare.14128991).

Phylogenetic analysis

We integrated the SOCS gene family sequences from different species and reptilian orders for phylogenetic analysis, and constructed an ML phylogenetic tree using RAxML with the full length of the SOCS gene sequences. Results showed a distinct and characteristic gene category classification pattern in which the orthologs clustered closer together compared to other closely-related members of the SOCS gene family (Fig. 1). SOCS gene family trees were rooted with the CISH gene, and their topologies were categorized into two groups, SOCS types I and II, indicating a high similarity across each classification. We also examined the phylogenetic relationship across SOCS genes and found that the evolutionary tree could be generally divided into four main clades: Serpentes, Sauria, Crocodilia, and Testudines (Fig. 2).

Figure 1 Phylogenetic tree of reptilian SOCS genes produced using RaxML.

The coding sequences of SOCS genes were used in the tree reconstruction. Trees of the SOCS gene family were rooted with the CISH gene, and their topologies were found to be categorized into two groups corresponding to the SOCS types I and II.

Figure 2 Phylogenetic tree of reptilian species and SOCS7.

Branches in blue, pink, green and red represent Serpentes, Sauria, Crocodilian and Testudines clades,respectively.

Identification of SOCS gene selection pressure

The sites’ selection pressure in their codon alignments were estimated by comparing the M7 and M8 models using the codeml program. Three site-based selection measures on the Datamonkey webserver (FEL, SLAC, FUBAR) were used to detect these sites. Considering each method’s randomness, only sites estimated by at least two methods were regarded as significant. FEL was effective at capturing rate variation and contained less false positive selection in small datasets (Pond & Frost, 2005). SLAC is a conservative method suitable for large data sets, but had flaws in its substitution rate estimation (Pond & Frost, 2005). FUBAR was faster at detecting positive selection and relaxed the restrictions of other models (Murrell et al., 2013). Positive signals in SOCS4 were detected by SLAC, FEL, and FUBAR methods in Datamonkey, indicating reliable adaptive evolution signals for SOCS4. However, positive signals were not strong for SOCS1, SOCS3, SOCS5, SOCS6, or SOCS7, and all the sites were only detected using one method (Table 1). We estimated that episodic selection or provisionally changed the bouts of selection in reptilian lineages using MEME on the Datamonkey web server at 0.05 significance. Reptilian phylogeny detection showed that SOCS4 possessed the maximum sites and underwent episodic selection, followed by SOCS5, SOCS1, CISH, SOCS2, and SOCS3. SOCS6 and SOCS7 showed no episodic selective sites.

Table 1 Tests for positive selection of SOCS genes in reptiles using site models.

Gene	No. of species	LnL M7	LnL M8	LRT P-value (M8 vs M7)	M8	SLAC	FEL	FUBAR	MEME	Total no. of sites	
CISH	30	−4,389.107568	−4,389.107643	0.999900005	0	0	38	0	218	0	
SCOS1	33	−4,345.285312	−4,345.28639	0.998900605	0	0	0	0	41,170	0	
SCOS2	29	−2,220.844129	−2,220.844683	0.999450151	0	0	126	123	126	0	
SCOS3	33	−3,289.79696	−3,289.25231	0.580044763	0	0	0	0	16	0	
SCOS4	37	−8,555.434409	−8,552.113207	0.036109474*	57	9,182	9,169,204	9	9,169,179,182,202,204	1	
SCOS5	38	−8,570.028335	−8,568.420091	0.200237724	0	0	38	0	63,67,92,123	0	
SCOS6	38	−7,176.04669	−7,175.868682	0.836942424	28,266,328	0	0	0	0	0	
SCOS7	22	−3,950.853562	−3,949.442767	0.243948047	1,104,316	0	0	0	0	0	
Notes:

* The significant level.

LnL means the log-likelihood score; the values in the columns M8, SLAC, FEL, FUBAR and MEME represent the positions of the amino acids.

The Z-test indicated that for each SOCS gene, the mean dN was smaller than the mean dS with a P-value < 0.01, implying significant purifying selection (Table S3). To further explore the prevailing selective action of the different reptile lineages across the eight SOCS members, we used the SOCS gene subsets Serpentes, Sauria, Crocodylia, and Testudines to estimate positive selection with branch and branch-site models in PAML. The branch model was performed by testing the one-ratio model (M0) vs. two-ratio model (M2) hypothesis in order to confirm lineage-specific adaptive events. The ω value (evolutionary rates) for all branches under M0 was less than 1, confirming that all SOCS genes underwent purifying selection, which was identical to the Z-test result. The M2 was implemented to examine whether the diverse foreground reptilian order lineages underwent dissimilar selection pressures compared to the background lineages. LRT indicated that M2 was more suitable for several lineages in each SOCS compared to M0 (P < 0.05) (Table 2). The ω values for the Serpentes lineages of CISH, SOCS4, and SOCS6; Sauria lineages of SOCS2, SOCS4, and SOCS7; Crocodylia lineage of SOCS3; and Testudines lineages of SOCS1, SOCS5, and SOCS7 were all significantly less than one, revealing that these SOCS genes had undergone forceful purifying selection. Furthermore, the Clade model C (CmC) was implemented to estimate if the Squamata, Crocodylia, and Testudines clades underwent different selection. The results showed that the SOCS genes (except SOCS3 (ωsquamata = 0.05807; ωCrocodylia = 0.36627; ωTestudines = 0.07415, P = 0.0270)) were not significantly better when compared to the M2a_rel model (P < 0.05) (Table 3). Because the branch model simply compares mean ω values for whole gene sequences rather than for one specific site, we used the branch-site model to estimate whether positive selection was acting on particular sites in different reptilian lineages. On the SOCS phylogenetic trees, we recognized several (foreground) branches with positive selection sites, but only the Crocodilian lineage in SOCS2 had significant levels (P < 0.01) (Table 4).

Table 2 Test for positive selection in divergent clades of SOCS genes with branch model.

Gene	Model compared	np	LnL	LRT P-value
(M2 vs. M0)	ω for branch	
CISH	M0	60	−4,442.284430		0.15453	
M2 (Serpentes)	61	−4,438.321911	0.004875635*	0.21853	
M2 (Sauria)	61	−4,441.416977	0.18778633	0.13084	
M2 (Crocodilian)	61	−4,441.908233	0.385717439	0.12301	
M2 (Testudines)	61	−4,441.512811	0.21414212	0.13356	
SOCS1	M0	66	−4,402.170189		0.12495	
M2 (Serpentes)	67	−4,401.956635	0.513415376	0.11473	
M2 (Sauria)	67	−4,400.650969	0.081316031	0.15359	
M2 (Crocodilian)	67	−4,400.810117	0.099091809	0.18927	
M2 (Testudines)	67	−4,399.94517	0.034901247*	0.06905	
SOCS2	M0	58	−2,239.918042		0.10314	
M2 (Serpentes)	59	−2,239.285271	0.260612577	0.12822	
M2 (Sauria)	59	−2,237.968878	0.048334995*	0.06648	
M2 (Crocodilian)	59	−2,239.909264	0.894458331	0.10882	
M2 (Testudines)	59	−2,239.210456	0.234195078	0.14166	
SOCS3	M0	66	−3,342.810516		0.0323	
M2 (Serpentes)	67	−3,342.677273	0.605689324	0.03601	
M2 (Sauria)	67	−3,341.527179	0.109135184	0.02591	
M2 (Crocodilian)	67	−3,338.338575	0.002783977*	0.09975	
M2 (Testudines)	67	−3,342.753792	0.736305998	0.03612	
SOCS4	M0	74	−8,769.926086		0.10547	
M2 (Serpentes)	75	−8,757.233899	0.00000047*	0.17336	
M2 (Sauria)	75	−8,757.160935	0.000000435*	0.07159	
M2 (Crocodilian)	75	−8,769.913378	0.8733744	0.10913	
M2 (Testudines)	75	−8,768.393871	0.080024702	0.14034	
SOCS5	M0	76	−8,709.270253		0.08233	
M2 (Serpentes)	77	−8,709.268176	0.948327344	0.08304	
M2 (Sauria)	77	−8,707.733022	0.079533938	0.0726	
M2 (Crocodilian)	77	−8,709.269578	0.970152858	0.08308	
M2 (Testudines)	77	−8,706.599832	0.020831911	0.11963	
SOCS6	M0	76	−7,228.208948		0.05398	
M2 (Serpentes)	77	−7,223.630666	0.002478226*	0.08087	
M2 (Sauria)	77	−7,226.13592	0.041730534*	0.04354	
M2 (Crocodilian)	77	−7,227.187074	0.1528375	0.03309	
M2 (Testudines)	77	−7,228.13578	0.702096588	0.05887	
SOCS7	M0	44	−3996.457239		0.04971	
M2 (Serpentes)	45	−3,993.427814	0.013836384*	0.07591	
M2 (Sauria)	45	−3,990.680729	0.0006764*	0.01896	
M2 (Crocodilian)	45	−3,995.805426	0.253556688	0.07603	
M2 (Testudines)	45	−3,994.495221	0.047601494*	0.08047	
Notes:

* The significant level.

M0 means the one-ratio model; M2 means the two-ratio model; LnL means the log-likelihood score; ω means the evolution rate.

Table 3 Test for positive selection in divergent clades of SOCS genes with Clade C model.

Gene	Model	np	LnL	k	Site class 0 (all branched)	Site class 1 (all branches)	Site class 2 (backgroud branches and different clade vary)	df	P-value	
CISH	M2a_rel (null)	63	−4,389.165103	3.55311	p0 = 0.67161	p1 = 0.01821	p2 = 0.31018			
					ω0 = 0.03353	ω1 = 1.00000	ω2 = 0.40891			
	CmC	66	−4,388.062024	3.55093	p0 = 0.69176	p1 = 0.00000	p2 = 0.30824	3	0.5307	
					ω0 = 0.03608	ω1 = 1.00000	ωClade0 = 0.28569			
							ωClade1 = 0.49948			
							ωClade2 = 0.38661			
							ωClade3 = 0.44745			
SOCS1	M2a_rel (null)	69	−4,342.291728	3.11579	p0 = 0.49785	p1 = 0.00000	p2 = 0.50215			
					ω0 = 0.01030	ω1 = 1.00000	ω2 = 0.25679			
	CmC	72	−4,338.801054	3.1002	p0 = 0.49008	p1 = 0.00000	p2 = 0.50992	3	0.0725	
					ω0 = 0.00925	ω1 = 1.00000	ωClade0 = 0.48684			
							ωClade1 = 0.25580			
							ωClade2 = 0.35633			
							ωClade3 = 0.13074			
SOCS2	M2a_rel (null)	61	−2,220.031054	3.53632	p0 = 0.62089	p1 = 0.00000	p2 = 0.37911			
					ω0 = 0.01055	ω1 = 1.00000	ω2 = 0.26823			
	CmC	64	−2,218.7223	3.55318	p0 = 0.67120	p1 = 0.00000	p2 = 0.32880			
					ω0 = 0.01768	ω1 = 1.00000	ωClade0 = 3.69064	3	0.4544	
							ωClade1 = 0.24540			
							ωClade2 = 0.33940			
							ωClade3 = 0.43711			
SOCS3	M2a_rel (null)	69	−3,289.766696	3.74457	p0 = 0.79659	p1 = 0.00758	p2 = 0.19583			
					ω0 = 0.00733	ω1 = 1.00000	ω2 = 0.13610			
	CmC	72	−3,285.178522	3.3514	p0 = 0.60843	p1 = 0.02774	p2 = 0.36383	3	0.0270*	
					ω0 = 0.00000	ω1 = 1.00000	ωClade0 = 0.68449			
							ωClade1 = 0.05807			
							ωClade2 = 0.36627			
							ωClade3 = 0.07415			
SOCS4	M2a_rel (null)	77	−8,550.47192	2.81933	p0 = 0.68919	p1 = 0.03621	p2 = 0.27460			
					ω0 = 0.01351	ω1 = 1.00000	ω2 = 0.27153			
	CmC	80	−8,550.38713	2.82065	p0 = 0.68778	p1 = 0.03632	p2 = 0.27590	3	0.9823	
					ω0 = 0.01336	ω1 = 1.00000	ωClade0 = 0.47495			
							ωClade1 = 0.26878			
							ωClade2 = 0.25278			
							ωClade3 = 0.29418			
SOCS5	M2a_rel (null)	79	−8,570.480518	2.93938	p0 = 0.76696	p1 = 0.01086	p2 = 0.22217			
					ω0 = 0.01332	ω1 = 1.00000	ω2 = 0.29751			
	CmC	82	−8,569.810117	2.93972	p0 = 0.75886	p1 = 0.01036	p2 = 0.23078	3	0.7195	
					ω0 = 0.01227	ω1 = 1.00000	ωClade0 = 0.27797			
							ωClade1 = 0.27905			
							ωClade2 = 0.29204			
							ωClade3 = 0.37277			
SOCS6	M2a_rel (null)	79	−7,175.477895	3.2453	p0 = 0.75850	p1 = 0.00567	p2 = 0.23583			
					ω0 = 0.00552	ω1 = 1.00000	ω2 = 0.19412			
	CmC	82	−7,172.895772	3.24907	p0 = 0.69726	p1 = 0.00703	p2 = 0.29572	3	0.1602	
					ω0 = 0.00000	ω1 = 1.00000	ωClade0 = 0.00000			
							ωClade1 = 0.17437			
							ωClade2 = 0.09228			
							ωClade3 = 0.19123			
SOCS7	M2a_rel (null)	47	−3,949.154154	3.3662	p0 = 0.79184	p1 = 0.01877	p2 = 0.18939			
					ω0 = 0.00000	ω1 = 1.00000	ω2 = 0.18827			
	CmC	50	−3,945.710868	3.38768	p0 = 0.79725	p1 = 0.01605	p2 = 0.18670	3	0.0756	
					ω0 = 0.00000	ω1 = 1.00000	ωClade0 = 0.07781			
							ωClade1 = 0.20122			
							ωClade2 = 0.27043			
							ωClade3 = 0.31754			
Notes:

* Significant level.

ωClade0 represents background clade; ωClade1 represents Squamata; ωClade2 represents Crocodylia; ωClade3 represents Testudins.

Table 4 Test for positive selection in divergent clades of reptilian SOCS genes by branch-site model.

Gene	Lineage	Models compared	np	LnL	P-value	Positively Selected Sites(BEB Analysis)	
CISH	Serpentes	Model A	63	−4,400.739546	1	22 S 0.808 40 E 0.843 218 S 0.789	
	Model A null	62	−4,400.739546			
Sauria	Model A	63	−4,403.555332	1	33 G 0.667 67 N 0.597 212 R 0.574 213 K 0.610	
	Model A null	62	−4,403.555332			
Crocodilian	Model A	63	−4,403.095376	0.476875571	79 L 0.889 226 Q 0.618	
	Model A null	62	−4,403.348368			
Testudines	Model A	63	−4,403.494378	1	79 L 0.763 180 V 0.727	
	Model A null	62	−4,403.494378			
SCOS1	Serpentes	Model A	69	−4,386.918212	1	145 S 0.766	
	Model A null	68	−4,386.918212			
Sauria	Model A	69	−4,378.882978	1	37 Q 0.614 86 P 0.500 130 K 0.754 175 F 0.996**	
	Model A null	68	−4,378.882978			
Crocodilian	Model A	69	−4,387.447863	1	41 D 0.596	
	Model A null	68	−4,387.447863			
Testudines	Model A	69	−4,387.964098	1	*	
	Model A null	68	−4,387.964098			
SOCS2	Serpentes	Model A	61	−2,232.642062	1	49 D 0.633 116 I 0.858	
	Model A null	60	−2,232.642062			
Sauria	Model A	61	−2,233.717143	1	NA	
	Model A null	60	−2,233.717143			
Crocodilian	Model A	61	−2,225.570868	0.000441506*	76 Y 0.502 145 V 0.994**	
	Model A null	60	−2,231.744746			
Testudines	Model A	61	−2,233.717143	1	NA	
	Model A null	60	−2,233.717143			
SOCS3	Serpentes	Model A	69	−3,317.167132	1	155 N 0.529 182 G 0.731	
	Model A null	68	−3,317.167131			
Sauria	Model A	69	−3,314.578678	1	NA	
	Model A null	68	−3,314.578678			
	Crocodilian	Model A	69	−3,300.34915	1	7 F 0.867 10 A 0.963* 12 M 0.961* 15 P 0.525 16 L 0.971* 28 K 0.992** 35 V 0.524 36 N 0.913 52 T 0.641 58 L 0.970* 103 S 0.643 130 H 0.545 171 L 0.840	
	Model A null	68	−3,300.34915			
Testudines	Model A	69	−3,317.168843	1	NA	
	Model A null	68	−3,317.168843			
SOCS4	Serpentes	Model A	77	−8,592.598731	1	50 E 0.757 53 S 0.991** 74 L 0.766 79 S 0.872 108 S 0.992** 113 V 0.880 175 A 0.764 176 S 0.933 179 G 0.968* 185 C 0.983* 191 C 0.780 205 N 0.597 231 K 0.736 237 E 0.788 389 A 0.880 418 E 0.543	
	Model A null	76	−8,592.598731			
Sauria	Model A	77	−8,606.251462	1	8 N 0.796 47 S 0.520 165 S 0.520 166 Q 0.613 171 D 0.985* 173 D 0.535 203 L 0.600 204 K 0.650 217 V 0.890 219 T 0.857	
	Model A null	76	−8,606.251463			
	Crocodilian	Model A	77	−8,615.179083	1	NA	
	Model A null	76	−8,615.179101			
Testudines	Model A	77	−8,614.992609	1	307 R 0.589	
	Model A null	76	−8,614.992608			
SOCS5	Serpentes	Model A	79	−8,610.143468	0.759898456	17 N 0.673 45 V 0.966* 54 S 0.696 87 T 0.857 123 K 0.756 331 S 0.539	
	Model A null	78	−8,610.190169			
Sauria	Model A	79	−8,609.864265	1	21 H 0.628 23 G 0.674 27 N 0.783 38 V 0.556 49 A 0.569 52 Q 0.923 72 T 0.877 86 V 0.653 121 N 0.650 153 V 0.693 172 M 0.938 235 L 0.564 291 L 0.864	
	Model A null	78	−8,609.864265			
Crocodilian	Model A	79	−8,614.616660	1	31 D 0.522 83 D 0.525 111 C 0.575 142 V 0.778 179 S 0.525 302 N 0.580 470 V 0.520 475 T 0.500 482 G 0.580	
	Model A null	78	−8,614.616660			
Testudines	Model A	79	−8,613.976725	1	93 Q 0.530 145 M 0.968* 178 Q 0.685 182 L 0.514	
	Model A null	78	−8,613.976725			
SOCS6	Serpentes	Model A	79	−7,194.275844	1	37 K 0.725 47 G 0.693 68 S 0.991** 94 V 0.878 248 V 0.802 271 V 0.655 279 V 0.873 321 N 0.524	
	Model A null	78	−7,194.275844			
Sauria	Model A	79	−7,204.549721	1	NA	
	Model A null	78	−7,204.549721			
Crocodilian	Model A	79	−7,204.549722	1	NA	
	Model A null	78	−7,204.549738			
Testudines	Model A	79	−7,204.549705	1	NA	
	Model A null	78	−7,204.549738			
SOCS7	Serpentes	Model A	47	−3,949.682893	1	67 A 0.862 72 N 0.806 92 S 0.834 285 A 0.878	
	Model A null	46	−3,949.682893			
Sauria	Model A	47	−3,958.004972	0.783490925	17 A 0.823	
	Model A null	46	−3,958.04274			
Crocodilian	Model A	47	−3,957.967949	1	NA	
	Model A null	46	−3,957.967949			
Testudines	Model A	47	−3,955.479092	0.612742908	108 H 0.553 116 Q 0.954*	
	Model A null	46	−3,955.607187			
Note:

LnL means the log-likelihood score; *represents the significant level; PPs of Bayes Empirical Bayes (BEB) analysis with P > 0.95 was regarded as candidates for selection (*>0.95, **>0.99).

Recombination and motif composition analysis

GARD was used to estimate putative recombination events, and the results showed no sequence exchanges or putative recombination events between the studied SOCS genes. The recombination results are shown in Table S3. The conserved SOCS protein motifs in reptiles were analyzed using MEME online software suite to detect the similarities and diversity in motif composition. We identified the conserved region across eight SOCS genes, and the results showed that all SOCS genes shared two of the same motifs (Fig. S1).

Discussion

An organism’s innate immunity acts as the first line of defense against infection (Dalpke et al., 2008). Over recent years, a series of studies on SOCS-deficient mice proved the significance of SOCS-mediated regulation of immunological and other crucial cellular responses (Banks et al., 2005; Lukasz et al., 2014; Metcalf et al., 2000; Naka et al., 1998). Previous studies also suggested that SOCS proteins are essential physiological regulators of both adaptive and innate immunity (Akihiko, Tetsuji & Masato, 2007). Reptiles share a common ancestor with mammals and hold an important amniote phylogeny position (Deakin & Ezaz, 2019). SOCS genes are important for reptile immune systems by enabling their adaptation to life in different environments. Although functional studies have explored the SOCS family’s crucial role in mammals, current knowledge of the SOCS gene repertoire in reptiles and its evolution is limited. The first reptile whole-genome sequence was the green anole lizard (Anolis carolensis) (Alföldi et al., 2011). The genomes of multiple species of turtles, snakes, lizards, and crocodiles have also been sequenced over the past decades, providing us with convenient conditions for analyzing the molecular evolution of reptiles using bioinformatics. In this study, we used genome-wide analysis to explore reptilian SOCS genes, as well as their phylogenetic relationship and adaptive evolution. To our knowledge, our study is the first comprehensive overview of the SOCS gene family within the reptilian genome.

Previous studies found that the SCOS family is comprised of eight members in mammals, and a second classification, type II (CISH, SOCS1–SOCS3), was added through two rounds of whole-genome duplication from a single precursor (Wang et al., 2019). In our study, we identified a total of 260 SOCS gene family sequences based on reptile genomic analyses. We identified eight intact SOCS family members in reptiles, the same number found in mammals, suggesting that the SOCS family expanded during the two rounds of whole-genome duplication (Dehal & Boore, 2005). This proved our hypothesis that reptile and mammal SOCS family members are conserved and also that reptiles and mammals shared similar common components in their immune systems (Zimmerman, 2020). We constructed an extensive phylogenetic tree from the SOCS gene coding sequences across the examined reptiles in order to analyze the evolutionary relationships. We found close relationships between SOCS4–SOCS7, which was consistent with previous studies (Linossi, Calleja & Nicholson, 2018). We examined the phylogenetic relationship across eight SOCS genes and found that the reptile SOCS genes could be classified into two groups: type I (SOCS4–SOCS7) and type II (CISH, SOCS1–SOCS3). When the SOCS7 gene tree was compared to the traditional species tree we found that genes’ phylogenetic proximity coincided with the morphological taxonomy (Fig. 2). The SOCS gene family evolved following the phylogeny of reptiles, confirming the crucial role of the SOCS genes. We performed motif identification using MEME analysis and found that diverse reptilian lineages shared high conservation in a two-motif structure of SOCS gene family members, which was consistent with the previously conserved SOCS gene structure (Hao & Sun, 2016). Previous studies confirmed that these two domains are necessary for the routine functions of the SOCS family (Fujimoto & Naka, 2003). These similarities indicate that SOCS genes are conserved in reptiles and mammals. Therefore, we considered that the SOCS gene sequences were highly conserved due to their essential role in regulating cytokine and growth factor signaling, and that reptile differentiation may be a significant driving force in the evolution of the reptilian SOCS gene family.

Positive selection is a crucial driving force in gene evolution in both function and structure, and the authentication of positive selection at the molecular level is important to the field of evolutionary biology (Vitti, Grossman & Sabeti, 2013). In this study, we focused on the selection test of SOCS genes on the whole reptilian phylogeny in order to estimate the adaptive evolution pressures acting on reptiles. Positive selection in SOCS4 was detected by at least two methods, indicating SOCS4’s strong adaptive evolutionary signals. No significant positive signals were detected in the remaining SOCS genes. SOCS4 has been reported to participate in HIF-1α regulation and acts as an adaptive mechanism to hypoxia (Kamura et al., 2000). Moreover, SOCS4’s convergent evolution among yak and Tibetan antelope was detected by a convergent signature and phylogenetic analysis, which might explain their high-altitude adaptations (Wang et al., 2015). Reptiles are ectotherms distributed across various environments such as marine, fresh-water, mountains, and flatlands (Zimmerman, Vogel & Bowden, 2010). Our results support the possibility of rapid SOCS4 evolutionary rates of reptiles when adapting to diverse environments.

Given reptiles’ wide distribution, we used branch model analyses to further explore whether positive selection acted on specific reptile lineages. The ω values for all of the target reptile branches were less than one, indicating that the primary force shaping SOCS gene evolution was purifying selection. Our study estimated the SOCS genes’ evolutionary tendencies under diverse selective pressures with the evolution of reptiles. SOCS genes in whole reptilian lineages maintained the protein structure in purifying selection (Mukherjee et al., 2009), and their crucial function acts as intracellular negative physiological regulators on cytokine and growth factor signaling (Croker, Kiu & Nicholson, 2008; Kershaw et al., 2013; Shuai & Liu, 2003). Although strong purifying selection pressures on SOCS genes have been detected in reptile lineages, the branch-site model analysis showed that two sites in the Crocodilian lineage on SOCS2 were under positive selection pressure. This demonstrated that there were discrepant selection pressures at different sites. SOCS2 is involved in cell growth and several inflammatory disorders (Letellier & Haan, 2016), and SOCS2 in Eriocheir sinensis has been shown to be associated with immune defense responses (Dalla Valle et al., 2009). SOCS2 has been found to undergo natural selection in Egyptian chickens when compared to Sri Lankan chickens, which might be due to the adaptation of Egyptian chickens to the arid, hot, and dry habitat (Walugembe et al., 2019). Moreover, SOCS2 has also been detected in selection signals of different high-altitude sheep, which might support Tibetan sheep when adapting to extreme environments of high altitude and ultraviolet radiation (Wei et al., 2016). These studies suggest that SOCS2 plays an essential role in the adaptation to a specific environment. Crocodilians are large, semiaquatic reptiles with strong tails and thecodont dentition used for hunting (Vickaryous & Gilbert, 2019). We speculate that the positive selection sites in Crocodilia might be linked with their semiaquatic habitat, which contains more environmental pathogens compared to that of their terrestrial relatives. However, the branch-site model analysis did not find any significant positive selection evidence for the remaining seven SOCS genes, which might be due strong purifying selection’s masking effect (Zhi-Yi et al., 2018). Differences in SOCS genes across reptiles may also reflect differences in selection pressure. The CmC model results suggested that SOCS3’s selection pressures were significantly different (P < 0.05) within divergent clades of reptiles, with different evolutionary rates identified in different lineages. This proves our hypothesis that SOCS family members are subject to different selection pressures in different reptilian clades. SOCS3 plays an essential role in modulating the outcomes of infections and autoimmune diseases by binding to both JAK kinase and cytokine receptors. Previous studies found that both the cetacean and reptilian TLRs evolved in response to environmental adaptations and rapid diversification (Shang et al., 2018; Shen et al., 2012). It has been shown that SOCS3 can be induced in innate immune cells, and SOCS proteins can act as direct inhibitors of TLR signaling, suggesting that they play an essential role in innate immunity (Baetz et al., 2004). Different reptile orders’ immune systems differ in terms of their evolutionary history, pathogen exposure, and other potential factors. They also vary greatly in terms of habitat, size, and life history (Pincheira-Donoso et al., 2013; Zimmerman, 2020). Thus, SOCS3’s diverse selection pressures in different reptilian clades may suggest a relationship between reptilian adaptation and a diverse living environment. Notably, we found that the evolutionary rate of SOCS3 in Crocodylia was extremely greater than that of Squamata and Testudines, suggesting that the SOCS3 in Crocodylia underwent rapid evolution. A recent study found that SOCS3 plays an important role in the regulation of glucose homeostasis during high-intensity exercise, which is necessary to maintain performance (Pedroso, Ramos-Lobo & Donato, 2019). Crocodilians are remarkably stealthy predators that stalk and ambush their prey (Erickson et al., 2012). We guessed that the rapid evolution of SOCS3 in crocodiles might be related to their unique predation mode, which requires greater energy at the moment of predation. However, all the clades showed similar ω values using the CmC model in the remaining seven SOCS genes, indicating identical selective pressures on these SOCS genes across different orders of reptiles. Despite the different lineages of reptiles acting on SOCS3, a strong purifying selection signature was detected across the eight SOCS members, which corresponded to the high sequence conservation in these genes. However, our speculations are based on evidence of selective pressure acting on sequence differences so additional functional studies are still needed to confirm our hypothesis.

Conclusion

This study is the first comprehensive analysis of SOCS gene family evolution in reptiles. A total of 260 SOCS sequences were identified in reptiles, and this detailed phylogenetic analysis offers a great basis for further functional studies. Eight intact SOCS family members were identified in reptiles, suggesting that the SOCS family has expanded during two rounds of whole-genome duplication. Our results suggest that SOCS genes are under purifying selection in reptiles, indicating that the SOCS gene family has stabilization and significant functional constraints. However, we identified evidence of positive selection in SOCS4 across reptiles, suggesting their adaptation to different types of habitats. Meanwhile, we determined that SOCS2 and SOCS3 had undergone rapid evolution in Crocodylia, which might be related to their environment and predation behavior. In summary, through multiple analysis and comparisons, we provided novel insights into the SOCS family’s molecular evolution in reptiles.

Supplemental Information

Supplemental Information 1 List of reptilian genome included in this study.

Click here for additional data file.

Supplemental Information 2 SOCS genes sequences information in reptiles.

Click here for additional data file.

Supplemental Information 3 Results of the Z-test of selection and GARD.

The “Test” column shows the results of dS-dN, and p < 0.01 indicates the significant negative selection; the “GARD” column shows the results of recombination.

Click here for additional data file.

Supplemental Information 4 Motifs locations of eight SOCS genes in reptiles.

Motifs in different color represent the conservative motifs; x-axis represent the position of amino acids.

Click here for additional data file.

Additional Information and Declarations

Competing Interests

Author Contributions

Data Availability

The authors declare that they have no competing interests.

Tian Xia conceived and designed the experiments, analyzed the data, prepared figures and/or tables, authored or reviewed drafts of the paper, and approved the final draft.

Lei Zhang conceived and designed the experiments, performed the experiments, analyzed the data, prepared figures and/or tables, and approved the final draft.

Guolei Sun conceived and designed the experiments, performed the experiments, authored or reviewed drafts of the paper, and approved the final draft.

Xiufeng Yang performed the experiments, authored or reviewed drafts of the paper, and approved the final draft.

Honghai Zhang analyzed the data, prepared figures and/or tables, and approved the final draft.

The following information was supplied regarding data availability:

The raw data are available in Table S2 and in figshare: Xia, T. (2021): SOCS gene family.phy. figshare. Dataset. DOI 10.6084/m9.figshare.14128991.v1.

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
