# Peer review of "Genomic evidence of adaptive evolution in the reptilian SOCS gene family"

_PeerJ, doi:10.7717/peerj.11677_

## Round 0.1 · original submission · Major Revisions

The manuscript poses an interesting question and addresses a problem that is likely to be well-read by a large number of researchers.

Both reviewers point out a large number of flaws in the manuscript, which in my opinion amount to a major need for revisions.

Importantly, both reviewers indicate that:

1. The manuscript requires better justification -- why was this system (reptiles) and biological feature (SOCS genes family) chosen? What makes this unique and interesting?

2. The methods require much more in-depth explanation and exposition. This means justifying the choices of analytical methods as well as fleshing out the multiple refinements made therein.

3. Results need a more clear presentation, especially figure labels.

4. Conclusions need to be more well-grounded to the results found.

There are further, very insightful comments made by both reviewers, and I suggest the authors take them all into account in preparing a revised version.

·

Basic reporting

The manuscript ‘Genomic evidence of adaptive evolution in the reptilian SOCS gene family’ proposes to elucidate the evolution of SOCS genes in reptiles. The strength of th epaper is that the authors implemented a diversity of methods to explore phylogenetic relations among genes, recombination and protein motif analysis. However, the relevance of using reptiles as a model system is not clearly justified, and this undermines the work.

Also, from what was found in studies with other vertebrates and SOCS genes (e.g., (Jin et al. 2008), I feel that the authors could have some clear hypotheses and predictions. Having those would make the paper stronger and facilitate understanding.

Finally, English is poor in several parts of the text and should be thoroughly revised.

Figures and tables need much more information in their captions.

Supplementary material should have legends.

Sequence data were deposited in public repository.

Experimental design

The research question is not well defined because the relevance for studying SOCS gene evolution in reptiles is not justified.

The authors used a variety of methods to test for genomic evidence of selection and recombination. Yet, given that the methods are not described in sufficient detail, the rationale of the analyses and the connection with the question is not clear.

The description of the analyses does not provide enough detail to be intelligible and repeatable.

Validity of the findings

As the paper is currently written, it is hard to judge the validity of the findings because the connections between the question and the design are not clear; a better description of the rationale behind the analyses needs to be provided; and results should be presented in a much clearer way.

Conclusions are not linked to original question because the question itself is not explicit.

Additional comments

I think the paper has potential because the authors explored a diversity of data on SOCS genes: phylogenetic analysis, recombination, motif analysis. However, the paper can be stronger if the relevance of using reptiles is clearly justfied and some hypotheses for SOCS early evolution are posed.
Also, the methods and results need to be better presented to enhance understanding and allow readers to judge the validity of the findings.

Below are my comments separated by sections.

Abstract

Line 49: It is not clear what ‘threat processes’ are. I suggest to clarify.
The knowledge gap – elucidating the origin of SOCS genes in reptiles – is not clearly linked to the design of the study. I recommend to make this link explicit, justifying why the deign and analyses you have done helps filling the knowledge gap.
The conclusion could be stronger if not focused on further studies, but focused on the main findings and making the implications to conservation more concrete.
Introduction
The connection between the two first paragraphs and the third, where reptiles are introduced to the story, needs to be made. Why are reptiles interesting to study in the context of SOCS genes evolution?
The question and the context to assess relevance are not clearly stated in the paper. From the title, I’m assuming the question is related to a greater understanding of SOCS gene evolution in early lineages of vertebrates. If so, that needs to be clearly stated in the text.
I suggest to move the sentences in lines 95-100 to the beginning of the third paragraph, but also to better state the relevance of SOCS gene evolution in reptiles. The description of reptiles provided in lines 90-95 does not clearly connect to SOCS gene evolution.

Material and Methods
Line 109: It is fine to place the detailed information about the reptilian genomes in the Supplement, however, some information of the phylogenetic scale (at least the major clades) that was studied is relevant and should be mentioned in the main text. This will allow readers to better assess the implications of this study.
Lines 117-119: Why is it relevant to perform such analysis?
Line 124: Once again, why perform a selective pressure analysis? If hypotheses are stated in the paper, it makes the connection between prediction and analysis much more concrete.

Evolutionary pressure analysis: Although there is quite detail on this analysis, it is very hard to follow what the tests actually do. I suggest starting with a general statement on how these tests work (general rationale) and the main differences between the models used.
Are the null models that the authors refer to testing whether evolution of SOCS genes is neutral? For the reader not completely familiarized with these methods, it is perhaps good to give some more basic information.
Lines 131-132: What are the models M7 and M8?
Recombination analysis and motif analysis: As for all the analyses done in the ms, I missed the justification for such analyses. What relevant information will these analyses provide? How do they connect to the question? Also, these methods should be more detailed to allow readers to understand and judge the validity of the analyses.

Results

Line 182: This statement is part of the discussion, and not of the results.
Lines 182-184: No information on class I and class II of SOCS genes appears in Figure 1, so the reader cannot verify the result. Also, these classifications should be defined in the methods.
Lines 184-185: Reptilian morphological taxonomy is not shown in Figure 1, so one cannot see the result stated.
Line 192: The limitations and reliability of the methods mentioned should be described in the methods then.
Lines 206-208: A reminder of what ω represents (and a description of it in the methods) will help to understand these results.
Lines 227-228: It is good to show the evidence for these ‘’negative’’ results, something that could give an idea to the readers of how the analysis measures recombination (or lack of it).
Lines 230-231: This sentence is not clear. Is the message that the same motifs were identified in all SOCS proteins?
Lines 233-234: If no citations are provided, one cannot verify if the motif results are indeed consistent with previous findings.
Figure 1 needs to be improved so that one can read the labels. And some more information of the reptilian clades should be provided (Testudines, Crocodilian, Sauria, Serpentes).
Figure 2 seems lacking information on the protein motifs. Given that the method used in this analysis is very poorly described, it is hard to grasp what the figure actually means. For instance, the different colors for each motif indicate codon sequences? All taxa have the same exact motifs?

Discussion

Lines 251-253: This finding is not explicit in the text or in any figure.
Lines 253-255: There is no evidence for this in the ms. How were gene trees and morphological taxonomy compared? What was measured? How was similarity tested?
Lines 275-277: How does this method show that selection pressures were different across clades? And different in what aspect? Mode of selection?
Lines 277-280: Even if it is true that selection pressures are different across species, it does not mean that the cause for this difference is the divergence in diet and habitats. These may be coincidental. Some kind of experimental manipulation would be needed to infer causality.

If results are better presented, then the discussion could focus on the implications of the findings, perhaps by comparing results within reptilian clades, and then comparing results with other vertebrates. Also, as the abstract suggests, what inferences could be drawn from the results to better understand the early evolution of SOCS genes?

Reviewer 2 ·

Basic reporting

The manuscript addresses an interesting topic of the SOCS gene family evolutionary history in reptiles. In this study, the authors conducted genomic comparative analysis including gene genealogy and evolutionary pressure analyses based on CDS extracted from deposited genomes, available to date. The main conclusion is that reptiles presented all SCOS genes and that the nature of selection pressure varies in different lineages of reptiles.

In general, the work has a clear, unambiguous, and professional English language used throughout.

Although database searches and phylogenetic analysis might be useful to get a broad overview of the distribution of members of the SCOS gene family, the present manuscript embraces several issues that prevent its publication as it is.

Thus, there are several points in the manuscript that need to clarify.

1. The introduction needs more detail and an improvement in the story-telling approach.
For example, in the first paragraph all sentences started very similar (and repetitive) - SOCS protein, SCOS gene family, SOCS family, and so on. I suggest rewriting these sentences in order to make them more compelling to ready.

2. In lines 75-76 the authors stated that the SOCS family of proteins can be classified based on the amino acid residues. These differences should be addressed properly.

3. Line 78 is very vague. Please mention at least some of the studies and diseases.

4. The link between paragraphs 3 and 4 are non-existent.

5. Line 90 states that "reptiles have a complicated and lengthy evolutionary history". This sentence sounds very awkward to me. The time is relative and depends on a referential - for those who study eukaryotes, 250 MA is quite recent. And the "complicated" was not defined.

6. There is no clear hypothesis.

7. Figures are relevant and of high quality. However, I think that they should be better labeled and described. I miss some crucial information as method (ML), rooted approach, and bootstrap values in Figures 1 and 2.

Experimental design

The bioinformatic approach is appropriate but needs more details. The methodology leaves many open questions that prevent the repeatability of the study.

1. Any reciprocal BLAST was applied?

2. The homology hypothesis was based solely on BLAST results or there were any additional confirmation steps?

3. There wasn't any information about the choice of GTR+T as the best evolutionary model. Any model selection software was used?

4. The reason for using CDS instead of protein sequences for the genealogy was not clarified in the text.

5. Lines 127-128 mentioned that a tree was downloaded from timetree (www.timetree.org) but was not clear which tree was that. I wonder if was a timetree for amniotes. If was, I think that it could be misleading - the newest MS used to build the timetree for reptiles was from 2016 and for amniotes was from 2013. Maybe there are better and more recent alternatives to check the phylogeny.

6. Lines 142-144 mentioned the performance of Z-test selection using JC model. Again, no model selection explicit thought the text.

7. I would like to see the raw data - at least the alignment - available elsewhere. I suggest the use of a repository - Figshare, Github.

Validity of the findings

1. Each gene was found in single or multiple copies in each genome? Is think it is a piece of relevant information to add.

2. Lines 185-187 mentioned that "phylogenetic trees were supported by reptilian morphological taxonomy". However it is not possible to test this information by looking at Figure 1. The name of the lineages are extremely small and there is no indication of bootstrap values.

3. Figure 1 - Is the tree rooted (mid-pointed?) or unrooted? This information is crucial and was not provided in the text.

4. Are the SOCS gene lineages well supported? I miss the bootstrap values.

5. Lines 236-242 are repetitions from the Introduction. Please rephrase.

6. I think that an aspect that could be better explored by the authors was how the genomic duplication events that occurred in vertebrates related to the evolutionary history of the SOCS gene family.

7. The discussion was very superficial and presented some circular thoughts. For instance, all selective force found was interpreted as a response to a diverse habitat and habits of reptiles. However, this diversification could explain any result found.
In my opinion, the authors should discuss their findings in the light of the different pathways involved in each SOCS members and possible physiological implications for the taxonomic lineages.

Additional comments

I commend the authors for their innovative idea and extensive bioinformatic work. In addition,
the manuscript is clearly written in a professional, unambiguous language. If there are some weakness, they are in the lack of compelling storytelling, a more detailed methodology, and the elaboration of strong conclusions (as I have noted above) which should be improved upon before acceptance.

---

## Round 0.2 · Minor Revisions

The modifications performed by the authors have been substantial and have improved the manuscript.

Reviewer #1 still has some important remarks. Please address all comments, especially:

1. more context in the importance of SOCS genes in reptiles
2. better comparison on the evolution and adaptation to environments with other related vertebrates

I agree with the reviewer that the English language requires some improvement still.

·

Basic reporting

English still needs some improvement.

Some information are still missing citations.

Figures are much clearer now and legends are adequate.
Raw gene sequences were shared.

Now the hypotheses have been stated, however more background on their justification needs to be provided.

The results are relevant for the hypotheses, but can be better explored in the discussion.

Experimental design

Original research aligned with the aims and scope of the journal.

Better definition of the question, but the knowledge gap still needs better justification.

Investigation was rigorous.

Methods are sound and now described in greater detail so to be understandable and reproducible.

Validity of the findings

Underlying data have been provided (gene sequences).

Conclusions are linked to the research questions, but the discussion is too speculative.

Additional comments

This version has improved in several aspects that were deficient in the previous version, especially in details of methods, in presenting hypotheses and in answering those hypotheses in the discussion.

However, I still think that more context identifying the relevance of studying Reptilian lineages should be presented in the introduction, for instance, mentioning in the introduction the relation between SOCS gene functions and habitat diversification. Also, why do we need a comprehensive review of the evolution of SOCS genes in reptiles? Just because it hasn't been done is not enough. I suggest the authors to talk about early evolution of SOCS genes and compare results with mammalian lineages.

Also, most of the discussion is speculative about the relations between different selection pressures in specific SOCS genes and different environments. I suggest the authors to connect their results with what was found in mammals. Were different selection regimes also found for SOCS genes in different lineages of mammals? If yes, what does that mean for interpreting the evolution of these genes.

Finally, English still needs revision.

---

## Round 0.3 · accepted · Accept

All comments by the reviewer were addressed, and the English language has been massively improved. Thank you for the thoughtful and careful revision of the manuscript.